# Evaluation of a Novel Dorsal-Cemented Technique for Atlantoaxial Stabilisation in 12 Dogs

**DOI:** 10.3390/life11101039

**Published:** 2021-10-02

**Authors:** Joana Tabanez, Rodrigo Gutierrez-Quintana, Adriana Kaczmarska, Roberto José-López, Veronica Gonzalo Nadal, Carina Rotter, Guillaume Leblond

**Affiliations:** 1Fitzpatrick Referrals Orthopaedic and Neurology, Surrey GU7 2QQ, UK; Joanat@fitzpatrickreferrals.co.uk (J.T.); Crotter@fitzpatrickreferrals.co.uk (C.R.); 2Small Animal Hospital, School of Veterinary Medicine, University of Glasgow, Bearsden G61 1QH, UK; Rodrigo.GutierrezQuintana@glasgow.ac.uk (R.G.-Q.); A.kaczmarska.1@research.gla.ac.uk (A.K.); roberto.joselopez.bcn@gmail.com (R.J.-L.); V.gonzalo-nadal.1@research.gla.ac.uk (V.G.N.)

**Keywords:** canine, spinal disorders, veterinary neurosurgery, atlantoaxial instability, craniocervical junction anomalies, rigid dorsal stabilisation, presurgical planning

## Abstract

Dorsal atlantoaxial stabilisation (DAAS) has mostly been described to treat atlantoaxial instability using low stiffness constructs in dogs. The aim of this study was to assess the feasibility and surgical outcome of a rigid cemented DAAS technique using bone corridors that have not previously been reported. The medical records of 12 consecutive dogs treated with DAAS were retrospectively reviewed. The method involved bi-cortical screws placed in at least four of eight available bone corridors, embedded in polymethylmethacrylate. Screw placement was graded according to their position and the degree of the breach from the intended bone corridor. All DAAS procedures were completed successfully. A total of 72 atlantoaxial screws were placed: of those, 51 (70.8%) were optimal, 17 (23.6%) were suboptimal, and 4 (5.6%) were graded as hazardous (including 2 minor breaches of the vertebral canal). Surgical outcome was assessed via a review of client questionnaires, neurological examination, and postoperative CT images. The clinical outcome was considered good to excellent in all but one case that displayed episodic discomfort despite the appropriate atlantoaxial reduction. A single construct failure was identified despite a positive clinical outcome. This study suggests the proposed DAAS is a viable alternative to ventral techniques. Prospective studies are required to accurately compare the complication and success rate of both approaches.

## 1. Introduction

Atlantoaxial instability (AAI) was first reported in dogs more than 50 years ago [1,2]. Geary et al. (1967) reported AAI in ten dogs of toy or miniature breeds; four of these dogs were managed surgically with a dorsal stabilisation technique using a simple wire loop [2]. Since then, various treatment options have been reported with ventral techniques becoming more popular over time, likely due to lower reported mortality rates [3,4].

AAI can occur subsequent to congenital, developmental, and/or traumatic causes [5,6]. Often minor trauma in dogs with pre-existing congenital anomalies can lead to subluxation [5]. Odontoid process malformation (aplasia or hypoplasia) is the most frequently reported cause of AAI in dogs [4]. Other associated atlantoaxial congenital anomalies include incomplete ossification of the atlas (C1), separation of the dens from the axis (C2), and insufficient ligamentous support [5,6,7,8]. These malformations are more commonly encountered in young toy breed dogs [4,5,6]. Regardless of the underlying aetiology, the dorsal displacement of C2 into the vertebral canal leads to neurological deficits and/or pain [5].

Various craniocervical junction anomalies can often complicate AAI cases, especially in young small and toy breed dogs [9,10]. These malformations include atlanto-occipital overlapping, C2 dens dysplasia, caudal occipital malformations, craniocervical junction dorsal fibrous bands, atlantoaxial incongruence, and occipito-atlantoaxial malformations (OAAM) [9,10,11]. The latter includes occipito-atlantal fusion (often unilateral), hypoplasia of C1 and/or C2 dens, various other C2 malformations and C1–C2 joint dysplasia with frequent features of AAI [12,13,14]. Atlantoaxial incongruence occurs when the size of C1 is disproportionate to that of C2 [15]. Recognising the complexity of these malformations is crucial to formulate an appropriate treatment plan and prognosis. It can be argued that complex craniocervical junction anomalies including those of C1–C2 incongruency and OAAM are more easily accessible via a dorsal approach [7,15,16]. 

Many surgical stabilisation techniques and conservative methods for the management of AAI have been reported [3,4]. Conservative management involving the use of cervical splints or bandages is often reserved for cases with subtle clinical signs, particularly small or skeletally immature dogs or where there are financial limitations [3,17]. Canine dorsal atlantoaxial stabilisation (DAAS) has mainly been described using low to moderate stiffness constructs such as orthopaedic wire, nonmetallic sutures, nuchal ligament, and Kirschner wires maintained with polymethylmethacrylate (PMMA) cement or a metallic tension band [5,18,19,20,21]. When compared with ventral stabilisation techniques, DAAS has been associated with higher mortality rates [3,4]. However, studies that report the outcome in dorsal techniques predominantly originate from several decades ago and most of them describe techniques that required penetration of the epidural space at the level of the C1 dorsal arch [2,18]. A modified ventral approach with either threaded pins or cortical screws embedded in cement has become the most reported technique achieving reasonably high success rates [3,22]. Reported success rates range from 50 to 94% with a trend towards a higher success rate in the past two decades [22,23,24,25,26]. Nevertheless, complications related to the ventral approach such as laryngeal paralysis, dyspnoea, dysphagia, or implant-failure-related complications are consistently reported [3,6,24,27,28,29].

To our knowledge, there is only one recent case series describing DAAS using screws and PMMA cement with a positive outcome reported in six dogs suffering from AAI and C1–C2 incongruence [15]. Jeffrey (1996) also reported a successful outcome in a Yorkshire Terrier following cross pinning of the spinous process of C2 to the wings of C1 and cement embedding [19]. The aim of this study was to assess the feasibility and surgical outcome of a modified rigid cemented DAAS. Here, we report a safe viable alternative to ventral techniques. This study also demonstrates that similar implant placement accuracy can be achieved from a dorsal approach when compared to ventral constructs [30].

## 2. Materials and Methods

### 2.1. Criteria for Case Selection and Data Collection

The medical records of all dogs treated with DAAS in two referral practices between 2019 and 2020 were retrospectively reviewed. The diagnosis of AAI was confirmed by advanced imaging and defined as appreciable dorsal displacement of C2 relative to C1 with/or without evidence of spinal cord compression or intramedullary lesions. Descriptive data collected for each dog included signalment, onset, and duration of clinical signs, neurological examination, dog video recordings, preoperative treatments, and postoperative notes including surgical complications. Using a modified Frankel scale [24], each dog was neurologically graded before surgery, on discharge and on short-term (<3 months postsurgery) and long-term (>6 months postsurgery) follow-up: grade 0 for normal gait without pain, grade 1 for normal gait with neck pain; grade 2 for proprioceptive ataxia; grade 3 for ambulatory tetraparesis; grade 4 for nonambulatory tetraparesis and grade 5 for tetraplegia.

### 2.2. Advanced Imaging

Diagnostic imaging, including MRI, CT, and plain radiography (if available), was reviewed by at least one board-certified neurologist. Magnetic resonance (MR) images were acquired under general anaesthesia using a 1.5 T scanner (either Tim system or Magnetom Essenza 1.5 MRI; Siemens AG, Erlangen, Germany). CT scan images were obtained either under sedation or general anaesthesia using either a 160-slice scanner (Aquillion PRIME Toshiba, Canon Medical Systems USA, Inc., United States) or a dual-slice scanner (Siemens Dual Slice Somatom Spirit, Siemens AG, Erlangen, Germany). CT images were used to evaluate the osseous structure for anomalous or traumatic lesions and for surgical planning. The images were imported and reviewed on Horos™ DICOM viewer, using bone window CT images in the 2D viewer, 3D multiplanar reconstruction, and 3D volume rendering modes.

### 2.3. Surgical Planning

Preoperative surgical planning was performed using 3D Slicer software (Surgical Planning Lab, Harvard Medical School, Harvard University, Boston, MA, USA, http://www.slicer.org, accessed on 27 September 2021). The optimal trajectory was determined in three planes by orientating screw models within the bone corridors of the 3D-reconstructed bone segmentation (Figure 1). C1 and C2 segments were realigned to an estimated anatomical location to facilitate visualisation of implant positions with respect to the sagittal plane. Subsequently, optimal screw diameters, screw entry points, inclinations between screw long axis and sagittal plane, and drilling depths of each implant were determined and exported to a Microsoft Excel^®^ spreadsheet (Microsoft Corporation, Redmond, WA, USA). Video recordings of the surgical plan including the 3D anatomy, entry points, and screw directions were generated for intraoperative visualisation (Appendix A). Purposed bone corridors included C1 lateral masses and wings (4 sites), C2 cranial articular surfaces, and cranial/caudal portions of C2 spinous process (4 sites). Occipital crest entry points were planned to avoid the transverse venous sinus where applicable. For selected cases, 3D printed drilling guides were printed with PLA filament using Ultimaker™ printer (Ultimaker, The Netherlands) and Ultimaker Cura software.

### 2.4. Surgical Technique

Dogs were positioned in sternal recumbency with slight elevation and ventral flexion of the head and secured to the table with tape and/or a vacuum cushion. A midline dorsal approach was performed from the occipital crest to the middle of the third cervical vertebra, elevating subcutaneous tissue and epaxial muscles until exposition of the dorsal surface of the atlantoaxial vertebrae was obtained. Gelpi retractors were carefully placed to allow gentle dissection around the cranial surface of C2 preserving C1 and C2 nerve roots. The stabilisation technique involved bi-cortical screws (stainless steel or titanium) placed in at least 4 of 8 available C1 and C2 bone corridors. The screw diameter was selected based on the surgical plan, ranging from 1.5 mm to 2.7 mm. Using a high-speed 1 mm burr, the entry points of each screw site were marked by burring through the cis cortex. Custom-made stainless-steel tubes were used over the drill bits to prevent tissue damage and to act as a drill stopper (Figure 2). Drilling direction was either estimated by visual assessment of a video recording depicting 3D screw positions on a computer display or using a wedge osteotomy gauge to match the calculated values of inclination angles to the sagittal plane (Figure 2c). When significant concerns were raised about occipito-atlantal instability, a titanium mesh and additional cortical screws were added to the construct extending it to the occipital crest. Depending on surgeons’ preferences and accessibility, partial articular surface drilling and bone allograft were performed within the C1–C2 synovial joint and between the C1 dorsal arch and C2 spinous process. Realignment of C1 and C2 was achieved by cautiously applying cranioventrally directed pressure on the spinous process of C2 whilst embedding the metal implants in polymethylmethacrylate cement. Routine multilayered suture followed. A postoperative CT scan was performed to determine screw placement quality.

### 2.5. Immediate Postoperative Care

All dogs received multimodal analgesia and supportive care whilst recovering from the procedure in the hospital environment.

### 2.6. Follow-Up

Short-term follow-up was assessed via physical examination, video footage, and client questionnaire. Long-term follow-up was obtained via physical examination, video, and/or telephone questionnaire. Where possible postoperative CT images were obtained at one of these time points or more. A successful outcome was defined as being ambulatory without reported or observed discomfort and without evidence of clinical deterioration when compared to prior to surgery. 

### 2.7. Implant Accuracy, Bone Fusion, and Implant Failure

Presurgical planned 3D screw position was compared with all available postoperative CT studies. Registration of the different time points was performed using 3D slicer software, aligning the bone anatomy and, if necessary, the PMMA cement (when significant bone growth occurred). Screws were classified as dangerous (vertebral canal violation equal or greater than ½ screw diameter), hazardous (vertebral canal violation less than ½ screw diameter or breaching intervertebral foramen or other unintended anatomical structures), suboptimal (including monocortical placement, breaching laterally of the bone corridor or inappropriate screw length) or optimal (bicortical and contained within the intended bone corridor) (Figure 3). A screw ratio rather than a metric measurement was used to account for the wide variation in dog size. Bone fusion and implant displacement were also subjectively assessed and recorded. All measurements and CT image analysis were performed by a single observer (G.L.).

## 3. Results

### 3.1. Signalment and Clinical Presentation

In total, 12 dogs with atlantoaxial instability (AAI) were included in this study (Table 1) with a mean age of 16.3 months (range 3.3–75 months) and a mean weight of 5.5 kg (range 1.5–25 kg). The most represented breed was the Chihuahua (n = 4), followed by Dachshund (n = 2). Two dogs were presented with chronic neurological signs—one with an acute on chronic presentation, three with acute/subacute presentations and four with paroxysmal episodes of presumptive pain and/or vestibular signs. Three dogs had a recognised traumatic event. Grade 4, nonambulatory tetraparetic dogs (33.3%) and grade 3, ambulatory tetraparetic dogs (25%) were the more frequent neurological grades encountered prior to surgery. Two of the four dogs with grade 4 tetraparesis had a sustained trauma and were classified as nonambulatory by both caregivers and referring veterinary surgeons and therefore examined in lateral recumbency. The only dog that presented tetraplegic had acutely deteriorated one day after attempted management of AAI using a dorsal suture technique with nonabsorbable sutures [31]. One case was treated with a cervical bandage for 3 months until a CT scan revealed significant worsening of the previous atlantoaxial luxation which prompted surgical intervention (case 2).

### 3.2. Preoperative Imaging Interpretation and Surgical Planning

All cases had a dorsal displacement of C2 causing spinal cord compression from either congenital or traumatic aetiology (Figure 4). Further relevant craniocervical findings are reported in Table 1. Intramedullary hyperintensity on T2-weighted images on the region of C1–2 was reported in four dogs (33.3%). Two dogs had displaced fractures; case 4 had a cranial C2 fracture and dorsal midline C1 fracture, and case 6 had a C2 body fracture through the cranial articular surfaces. Atlantoaxial incongruence was present in 33.3% of the cases (n = 4), atlanto-occipital overlap was encountered in two dogs (16.6%) and two dogs had complex occipito-atlantoaxial malformations with partial atlanto-occipital fusion (16.6%). Dens hypoplasia was present in four dogs, while dens aplasia was present in two dogs. Other concomitant findings were occasionally reported on MRI including ventriculomegaly in 5 dogs (41.7%) and supracollicular fluid accumulation in three dogs (25%). Of the three dogs with a traumatic injury, two had preexisting congenital malformations (dog 4 had an incomplete fusion of C1 ventral arch, and dog 10 had a complex OAAM).

Preoperative surgical planning was used in all but one dog (case 4) and 3D-printed guides were used in the first two dogs. Some neurosurgeons preferred to use visual assessment of screw directions (R.G.Q. and R.J.L.), while others preferred to use numerical values and osteotomy wedge gauge (G.L.).

### 3.3. Immediate Surgical Outcome

All DAAS procedures were successfully completed allowing stabilisation of C1–C2 and occasionally also involving the occipital bone due to partial occipito-atlantal fusion (Figure 5). The median surgery time was 207 min (range 155–300 min). No major complications were reported intraoperatively. One case had CSF leakage following a small incision through the lateral aspect of the vertebral canal within the intervertebral foramen. Mild-to-moderate haemorrhage was commonly observed around the lateral foramen and C1–C2 intervertebral foramen often prolonging dissection time. On recovery, one dog had a few episodes of regurgitation immediately postoperatively which responded to proton pump inhibitor treatment (omeprazole, 1 mg/kg/12 h).

Based on postoperative CT images, apposition was considered optimal in all cases. A total of 72 atlantoaxial screws were placed (Table 2), with 51 (70.8%) graded as optimal. Four screws (5.6%) located in C1 lateral masses (n = 2) and C2 cranial articular surface (n = 2) were graded as hazardous—two had minor vertebral canal breach, one was excessively long, and one breached the alar foramen. The remaining 17 (23.6%) screws were considered safe but not perfectly placed within the intended corridors, most commonly due to a monocortical position in 11 (15.3%) screws. None of the screws were graded as dangerous. A titanium mesh affixed to the occipital bone was added to the construct in one dog suffering from a complex occipito-atlantoaxial malformation (Figure 5c). A case example is presented in Figure 6, depicting C1–C2 reduction and screw accuracy.

### 3.4. Perioperative Outcome

Mean time to discharge after surgery was 3.3 days (range 2–10 days). The dog with the longest hospitalisation time had a C2 vertebral body fracture and presented with nonambulatory tetraparesis. All dogs except one were discharged with one or more of the following medications: an NSAID, prednisolone at an anti-inflammatory dose, paracetamol, and/or gabapentin. At discharge, 6/12 dogs were graded as unchanged, compared to admission, 4/12 improved by at least one grade, and 2/12 were considered worse. Two dogs were discharged with subtle torticollis, one of which later developed a subcutaneous seroma in the surgical region.

### 3.5. Short-Term Clinical Outcome

In total, 11 dogs were reexamined between 1 to 2.8 months after surgery (mean 2 months). Neurological grading was performed for each dog (Table 1). All owners reported improvement in gait and/or painful episodes. One dog was reported to be unable to jump despite his gait being much improved (case 1), and one was reported to have rare episodes of yelping and stiffness (case 2). 

Case 7 was reported to have two further episodes of syncope or seizures and to experience reverse sneezing. Three episodes of dysphagia were reported after swallowing entire biscuits. Repeat MRI, CT, CSF analysis, echocardiography, and a continuous ECG (Holter monitoring) failed to identify a cause. Vagal syncope or seizure-like episodes were the main differential diagnoses. The latter hypothesis could be related to ventriculomegaly and supracollicular fluid accumulation. Frequent neck scratching was also reported in this case which was considered secondary to the identified Chiari-like malformation. Omeprazole (10 mg/kg BID) and gabapentin (17 mg/kg, BID) were prescribed. 

Case 8 was improving until an acute deterioration 4.9 months after surgery. Dynamic radiographs and CT imaging suggested the presence of atlanto-occipital instability which worsened with ventroflexion of the head. This dog was managed medically with rest and a neck brace. 

Case 9 developed paroxysmal episodes that were vestibular in nature (vomiting, horizontal nystagmus, vestibular ataxia) that lasted hours and then returned to normal. A repeated MRI scan, CSF analysis, bile acid stimulation test, and full body CT scan failed to identify a cause. The dog was managed with a hypoallergenic diet and gabapentin.

### 3.6. Long-Term Clinical Outcome

Long-term follow-up questionnaires were obtained in 11/12 dogs, while long-term neurological examination and CT scan were obtained in 10/12 dogs at a mean time of 11.9 months (5.9–19.8 months). All owners were satisfied with the clinical outcome and reported improvement with the gait and/or painful episodes. All dogs had a good-to-excellent outcome and had improved neurologically by one or more grades (Table 1). 

Case 2 developed rare episodes of suspected pain and weakness of the pelvic limbs 20 months after surgery. Further investigation was declined, but these clinical signs were considered less likely to be related to the atlantoaxial surgery given the suspected location of discomfort although an association cannot be completely excluded. The owner also reported very occasional coughing when drinking water. 

Case 3 presented rare episodes of 2–3 seconds collapse of the thoracic limbs along with paddling without autonomic signs which could be related to persistent AAI given that construct failure was identified on CT images. 

Case 8 was treated with a neck brace for 8 weeks following an acute deterioration. The dog improved neurologically and was only slightly ataxic in all four limbs with a subtle hypermetric gait. A right pelvic limb lameness was also reported which was attributed to medial patellar luxation. 

Case 7 experienced another seizure-like episode 7 months after the previous. This dog continued to be treated with omeprazole and gabapentin and was doing clinically well in between paroxysmal episodes. Previously reported neck scratching had resolved at long-term follow-up. 

Case 9 had a decreased frequency of episodes of vestibular nature. A hypoallergenic diet was started, and no further episodes had been observed for 3 months (at the time of long-term follow-up). The caregivers decided not to pursue any further therapeutical trials given that the dog had otherwise a good quality of life and was able to exercise normally.

### 3.7. Bone Fusion and Implant Failure

Overall, stabilisation constructs were considered appropriate and withstood the follow-up period in all cases except one (Table 2). The construct failure was attributed to poor cement embedding of the right lateral mass screw, leaving only one C1 screw supporting the construct which was subsequently pulled out. This implant failure led to appreciable C1–2 subluxation on follow-up CT; however, the dog still had a positive clinical outcome.

Bone allograft was placed in four cases, with only one having signs of C1–C2 fusion dorsally and bilaterally on CT scan after 19.3 months after surgery. Both cases with displaced fractures had signs of fusion of the fracture line, one also displayed C1–C2 fusion dorsally while the other displayed signs of bone remodelling of C1 dorsal arch and C1–C2 articular surfaces without complete fusion (residual bone separation line). Two other dogs had signs of bone remodelling ventrally. Overall, continuous C1–C2 bone fusion was observed in two dogs (16.6%), and partial bone fusion (with residual separation line) in three dogs (25%). Significant bone growth could be appreciated in younger dogs when overlapping the presurgical CT images with the long-term ones. Growth mostly occurred within the C1 wings, the caudal portion of the C2 vertebral body, and C2 transverse processes.

## 4. Discussion

The present case series suggests that DAAS cemented constructs can be safely performed to treat a wide range of craniocervical junction anomalies and traumatic injuries. All the dogs improved clinically during the initial follow-up period, and only two dogs showed mild short-lived clinical deterioration in the immediate postoperative period. No mortality event has been recorded at the time of writing. Whilst the number of cases reported in this series is modest, our data still suggest that perioperative mortality associated with rigid DAAS is likely to be low. The limited published data on rigid DAAS cemented constructs also suggest a low mortality rate [15,19]. Previous retrospective studies have associated dorsal techniques with higher mortality rates [3,4]. However, these conclusions were based on procedures which involved penetration of the vertebral canal at the level of C1 dorsal arch and were, therefore, more susceptible to iatrogenic trauma of the spinal cord [2]. Overall, if our results can be reproduced on a larger scale, we anticipate that success rates of rigid DAAS will likely be similar to that of ventral techniques.

Defining the parameters of a successful outcome can be difficult in veterinary medicine. Most studies define surgical success in terms of improved subjective gait scoring and absence of discomfort. Based on such criteria, 11 of our 12 dogs would be considered successful. Yet, our case series also highlights the complexity of AAI cases; many dogs had comorbidities that significantly affected the quality of life and therefore outcome. Further complicating the assessment of outcome, it is possible that residual C1–C2 instability may be intermittent or even subclinical, and therefore, construct failure may initially go undetected. In this series, three of our dogs suffered from paroxysmal episodes at long-term follow-up which may or may not be related to the DAAS procedure. One dog (case 8) acutely deteriorated 4.9 months after surgery and dynamic imaging revealed the presence of atlanto-occipital instability without any evidence of construct failure. Another dog (case 3) displayed a positive clinical outcome, yet follow-up CT images revealed that the supporting implants were failing. These cases illustrate the fact that success is difficult to reduce to a single objective criterion in dogs suffering from AAI. When comparing AAI studies, it may, therefore, be more relevant to compare mortality/complication rates and technical surgical outcome variables rather than clinical scoring.

A significant aspect of spinal instrumentation procedural safety is related to the surgeon’s ability to position stabilising implants accurately within the intended bone corridors and away from vital structures. Our data suggest that the proposed method can achieve a high level of screw placement accuracy with only four hazardous screws (5.6%) and no dangerous screws identified. This result is comparatively superior to two separate cadaveric and clinical studies assessing implant placement accuracy using ventral cemented techniques (4.4% dangerous screws) but similar to a ventral technique using 3D-printed drill guides (7% incomplete vertebral canal breach) [28,30,32]. It is difficult to know to which extent our preoperative planning methodology optimised screw placement accuracy. Anecdotally, the single case in which the planning method could not be used had the poorest screw placement scores, including two of the four identified dangerous screws in our entire population. Further investigation would be necessary to quantify the effect of preoperative planning on screw placement accuracy. The most challenging bone corridors proved to be the C1 lateral masses and C2 cranial articular surfaces. This could have been anticipated considering these corridors have a narrower shape. In theory, these two bone corridors could be completely avoided by only using the C1 wings and C2 spinous process, as was previously reported in a single case report [19]. However, the remaining bone corridors are extremely thin and may not be sufficient to sustain long-term cyclic loads. The optimal number and distribution of screws remains to be established for both ventral and dorsal techniques. In the absence of comparative data, our strategy was to use as many bone corridors as possible to maximise the construct’s bone anchorage. As confidence in the technique and surgical approach increased, it became possible to achieve screw placement in all eight available corridors. To our knowledge, the use of the C2 cranial articular surface corridors has not been previously reported from a dorsal approach. Based on our experience, this corridor can be technically challenging in smaller dogs, but it offers significantly more bone reserve than the C2 spinous process and is particularly valuable to stabilise cranially located C2 fractures.

One of the hypothesised benefits of the dorsal approach is that it may prevent complications that have been historically attributed to iatrogenic injury of ventrally located vital structures such as the vagosympathetic trunk, larynx, or oesophagus. Such injuries have been suspected in several ventral atlantoaxial stabilisation studies reporting postoperative complications such as Horner’s syndrome, dysphagia, dysphonia, laryngeal paralysis, dyspnoea, aspiration pneumonia, or tracheal necrosis [2,6,22,24,27,28]. While we have not directly observed such clinical signs, two of our cases suffer from mild dysphagia (cases 2 and 7). In humans, postoperative dysphagia and regurgitation following an anterior approach have been linked to a craniocervical malalignment in a hyper-flexed position which results in narrowing of the oropharyngeal space [33]. Such misalignment could, in theory, also occur from a dorsal approach, and therefore, we cannot exclude that the mild reported dysphagia may be a consequence of the DAAS surgery. Large comparative studies would be needed to properly establish any potential benefit/detriment of the chosen surgical approach on such infrequent complications.

Another benefit of the dorsal approach is that it offers access to the caudal occipital bone and C1–C2 dorsal laminae, allowing surgical interventions such as occipito-atlantal stabilisation and/or dorsal decompressive craniotomy or laminectomy [12,15,16,34,35,36,37]. This study demonstrated that craniocervical stabilisation could be successfully implemented even in the presence of complex OAAM with partial occipito-atlantal fusion (cases 9 and 12). Such anomalies are expected to cause an exacerbated fulcrum effect on the atlantoaxial joint, and it is, therefore, important to optimise the biomechanical properties of the associated stabilising construct [12]. Anatomically, the dorsal approach provides biomechanical advantages in that it allows the placement of the implants along the tension surface of the vertebral column [38,39]. It also offers opportunities to extend the position of stabilising implants further rostrally than would be possible with the ventral approach using a titanium mesh affixed to the cranium. Most of the available literature associated with atlanto-occipital instability describes ventral stabilisation techniques, with stabilising constructs often limited to the atlantoaxial region. These methods generally achieved a successful outcome, but several implant failures have also been reported [12,16,34,35,36]. To our knowledge, cases 9 and 12 represent original reports of complex OAAM solely treated via dorsal stabilisation. 

Atlantoaxial incongruence and dorsal fibrous bands are further examples of lesions that can benefit from a dorsal approach [15,34]. Our results, along with a previous single study, support rigid DAAS as an effective and safe method for the management of C1–C2 incongruence [15]. Based on our experience, DAAS allows partial resection of the C2 dorsal lamina/spinous process when surgical reduction of a hypoplastic C2 causes dorsal spinal cord compression. It has also been previously argued that semirigid dorsal constructs would not be biomechanically appropriate without proper C1–C2 joint congruency [15]. 

Based on our experience, the main limitation of the proposed DAAS is the limited access to the C1–C2 synovial joint for bone grafting (dorsolateral extremities) [5]. Despite using bone grafting in four cases, only two cases demonstrated convincing C1–C2 fusion on long-term CT images, and three dogs achieved partial bone remodelling consistent with significant ankylosis. A ventral approach may offer higher arthrodesis potential considering that the articular surfaces are readily accessible, although fusion rates have not yet been established to our knowledge. Another technical difficulty associated with the proposed DAAS method is the delicate dissection required, in particular around the lateral and intervertebral foramen. Finally, the most significant limitations of this study are the small population size and the retrospective design. 

## 5. Conclusions

This study suggests the proposed DAAS is a viable alternative to ventral techniques and can be safely used to treat a variety of craniocervical junction disorders. We believe that this technique has the potential to reduce complication rates related to the disruption of vital anatomical structures located ventrally. Prospective studies would be necessary to accurately compare complication and success rates of DAAS to that of a ventral technique. Further investigation into the role of preoperative planning and determination of the optimal number of stabilising cortical screws would also be beneficial. 

## Figures and Tables

**Figure 1 life-11-01039-f001:**
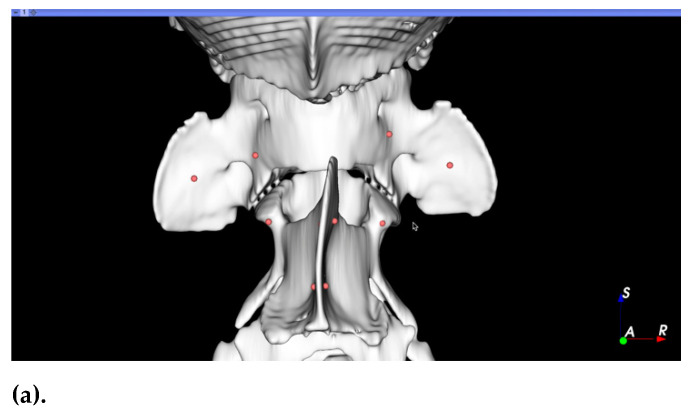
Example of surgical planning screenshots used to guide screw placement intraoperatively: (**a**) dorsal view of C1–C2 with planned screw insertion points depicted in red; (**b**) left lateral view used to estimate the approximate orientation of the planned screws in a craniocaudal direction; (**c**) caudal view of C1–C2 used to visually estimate the screw inclination with respect to the sagittal plane which could also be obtained using numerical values depicted in the associated table. Appendix A further demonstrates how these images can be used for anatomical, screw insertion, and screw orientation visualisations.

**Figure 2 life-11-01039-f002:**
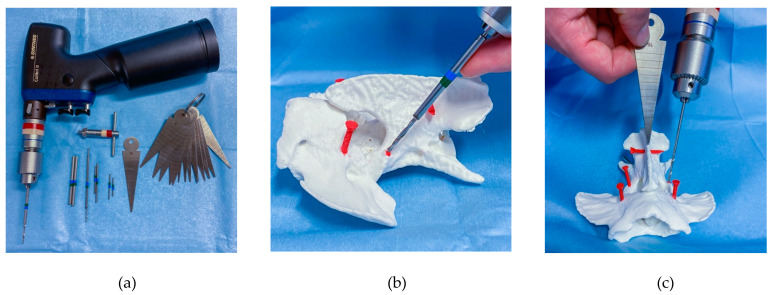
Photographs demonstrating the use of a drill stopper and inclination guide on a 3D printed C1–C2 model: (**a**) instruments used to accurately drill through the bone corridors including a surgical power drill, drill bits, custom-made drill stoppers, and osteotomy wedge gauge; (**b**) a drill stopper in the form a stainless steel tube (marked with green and blue tape) can be used to protect adjacent tissue and control drilling depth, the craniocaudal drilling inclination is guided by 3D planning images (compare to green screw axis in Figure 1b); (**c**) osteotomy wedge gauge are simple stainless steel triangles that can be used to approximate the inclination of drilling with respect to the sagittal plane, the angle values for each screw site were calculated and used at the surgeon’s discretion (see Figure 1c).

**Figure 3 life-11-01039-f003:**
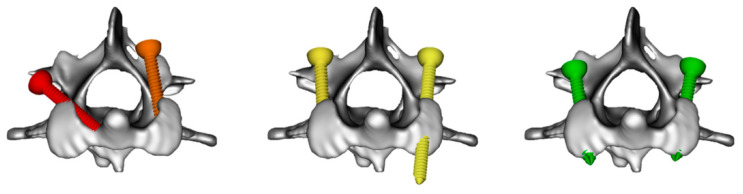
Axis 3D reconstructions depicting our proposed screw position grading system. Examples include a dangerous screw with vertebral canal violation greater than ½ the screw diameter (red), a hazardous screw with vertebral canal violation less than ½ the screw diameter (orange), two suboptimal screws (yellow) including a monocortical placement (**left**), and an inappropriate length (**right**) and two optimal screws (green).

**Figure 4 life-11-01039-f004:**
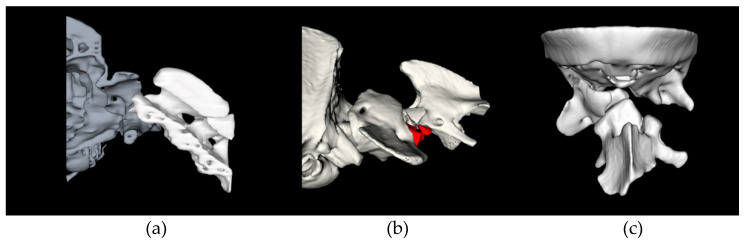
Craniocervical 3D reconstructions depicting the range of anomalies treated in this study: (**a**) congenital atlantoaxial instability; (**b**) C2 fracture; (**c**) complex occipito-atlantoaxial malformation.

**Figure 5 life-11-01039-f005:**
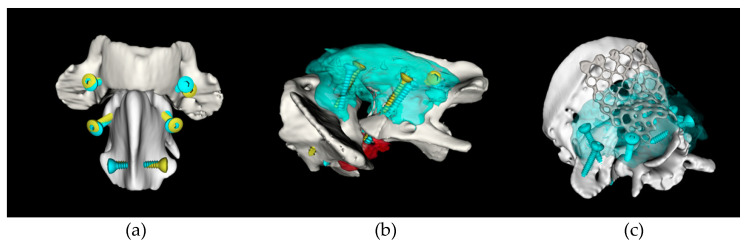
Craniocervical 3D reconstructions depicting the range of stabilisation constructs and screw accuracy obtained with the proposed method: (**a**) dorsal view of a construct with screws in C1 lateral masses (n = 2), C2 cranial articular surfaces (n = 2), and C2 spinous process (n = 1) used to stabilise a congenital AAI; (**b**) left lateral view of a similar construct which can also be used to treat cranial C2 fractures; (**c**) dorsal oblique view of a complex construct involving screws placed in all 8 available bone corridors and a titanium mesh affixed to the occipital crest. Yellow screws: preoperatively planned position; blue screws: actual postoperative position; blue semitransparent areas: polymethylmethacrylate cement location.

**Figure 6 life-11-01039-f006:**
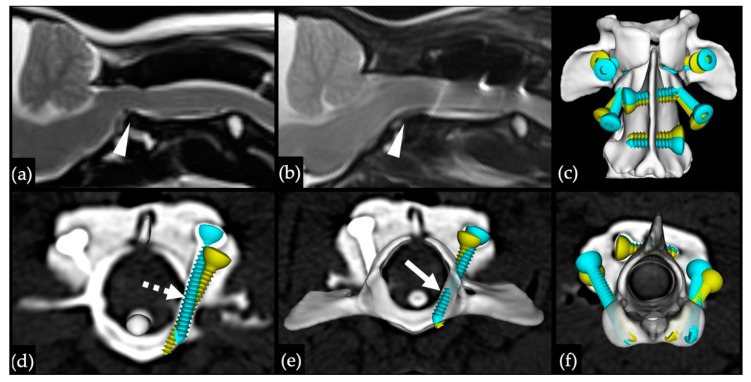
MR and CT images with superimposed 3D models depicting C1–C2 reduction and screw accuracy obtained in case 7: (**a**) preoperative sagittal MR image with dorsal displacement of C2 dens (white arrowhead); (**b**) postoperative sagittal MR image demonstrating C1–C2 reduction and the benefit of using titanium screws allowing visualisation of the local anatomy; (**c**) dorsal 3D reconstruction depicting the overall screw placement accuracy; (**d**) cranial view of CT transverse image illustrating a hazardous position of the left lateral mass screw due to slight vertebral canal violation (dashed arrow); (**e**) caudal view of a CT transverse image and C1 3D model revealing the suboptimal position of the right lateral mass screw due to penetration of the C1–C2 synovial joint (white arrow); (**f**) cranial view of a CT transverse image and C2 3D model depicting 3 optimal screw placements (bicortical within the intended bone corridor). Yellow screws: preoperatively planned positions; blue screws: actual postoperative positions.

**Table 1 life-11-01039-t001:** Clinical information, complications, and outcomes.

Case	Signalment (Breed, Age in Months, Gender, Weight)	Onset/Clinical Progression	Additional Anomalies	Surgical Time (Sx), Hospitalisation (H) and Complications (C)	Neurological Score ^1^		Final Outcome (Time in Months)
					Initial	Follow-Up	
1	Dachshund, 4 m, ME, 3.3 kg	Chronic and progressive	Dens hypoplasia, C1–C2 incongruence	Sx: 260 min H: 2 days C: none	Ad: 3 Dis: 3	ST: 3 LT: 2	19 m: improved gait, unable to jump
2	Maltese, 17 m, ME, 1.5 kg	Paroxysmal episodes pain	None	Sx: 155 min H: 2 days C: regurgitation (short lived)	Ad: 1 Dis: 2	ST: 2 LT: 0	20 m: much-improved gait; rare episodes of mild pain, occasional cough when drinking
3	Chihuahua, 10 m, FE, 2 kg	Chronic progressive with peracute deterioration	Dens hypoplasia, atlanto-occipital overlap	Sx: 300 min H: 4 days C: subtle torticollis.	Ad: 4 Dis: 4	ST: 3 LT: 2	11 m: improved gait; rare episodes of 2–3 seconds thoracic limb collapse and limb paddling.
4	Labrador Retriever cross, 3 m, FE, 9 kg	Trauma C1 and C2 fracture	Incomplete fusion of C1 ventral arch	Sx: 215 min H: 3 days C: none	Ad: 4* Dis: 3	ST: 0 LT: 0	12 m: normal gait; subtle stiffness of the neck.
5	Chihuahua cross, 23 m, FN, 2.2 kg	Chronicprogressive	Dens aplasia, C1–C2 incongruence	Sx: 165 min H: 4 days C: none	Ad: 3 Dis: 3	ST: X LT: 0	16 m: normal gait
6	German Shorthaired Pointer, 6 m, ME, 25 kg	Trauma C2 fracture	None	Sx: 175 min H: 10 days C: none	Ad: 4* Dis: 2	ST: 0 LT: 0	16 m: normal gait
7	Chihuahua, 11 m, MN, 2.2 kg	Single acute episode of pain, collapse, and ataxia	C1 incomplete dorsal arch fusion, caudal occipital malformation,	Sx: 220 min H: 2 days C: none	Ad: 2 Dis: 2	ST: 2 LT: 0	10 m: improved gait; 3 further paroxysmal episodes of syncope vs. seizures. Reverse sneezing and occasional dysphagia. Frequent neck scratching responding to gabapentin
8	Yorkshire Terrier, 4 m, FE, 1.7 kg	Subacute	Dens agenesis, atlanto-occipital overlap (suspect instability)	Sx: 160 min H: 2 days C: none	Ad: 5 Dis: 4	ST: 2 LT: 2	8 m: improved gait; mild ataxia in all four limbs (suspected occipito-atlantal instability 5 m postsurgery due to acute deterioration)
9	Cockapoo, 4 m, FE, 3.8 kg	Acute after collision with another dog	Complex OAAM, partial occipito-atlantal fusion, C1–C2 incongruence	Sx: 220 min H: 4 days C: none	Ad: 4 Dis: 2	ST: 0 LT: 0	6 m: improved gait; subtle low head carriage.Rare paroxysmal vestibular episodes (improving with diet adjustment/gabapentin)
10	Chihuahua, 34 m, MN, 2.1 kg	Paroxysmal episodes of pain and lateral recumbency	C1–C2 incongruence	Sx: 165 min H: 2 days C: none	Ad: 1 Dis: 2	ST: 0 LT: 0	6 m: return to normal (phone communication only)
11	Dachshund, 75 m, FN, 4.7 kg	Paroxysmal vestibular episodes ^2^	C2–3 block vertebrae	Sx: 245 min H: 3 days C: none	Ad: 0 Dis: 0	ST: 0 LT: 0	6 m: normal gait. No episodes since surgery
12	Lagotto Romagnolo, 4 m, FE, 8.5 kg	Acute and progressive	Complex OAAM, partial occipito-atlantal fusion	Sx: 200 min H: 2 days C: subcutaneous seroma and subtle torticollis	Ad: 3 Dis: 3	ST: 1 LT: 0	6 m: improved gait; mild over-reaching of all four limbs. Slight resistance to cervical ventroflexion

^1^ Neurological grades using a modified Frankel scale: 0, normal gait without neck pain: 1, normal gait with neck pain; 2, proprioceptive ataxia; 3, ambulatory tetraparesis; 4, nonambulatory tetraparesis; 5, tetraplegia [24]. Ad: admission; Dis: discharge; ST: short term; LT: long term, X: not available; *: assessed in lateral recumbency. ^2^ Reproducible paroxysmal episodes of vestibular signs elicited with flexion of the head (evaluated via video recording provided at the time of referral). ME: male entire; MN: male neutered; FE: female entire; FN: female neutered.

**Table 2 life-11-01039-t002:** Accuracy of screw placement and long-term construct analysis.

Case	Screw Sites (n)		Screws Grading Score ^1^		Bone Graft	C1–C2 Fusion (Location)	Construct Failure	CT Time Postsurgery (Months)
	C1	C2	n	Sites				
1	LM: 2 Wi: 0	AS: 2 SP: 1	Score 0: 4 Score 1: 1 Score 2: 0	LM, AS, SP LM	Yes	Yes (C1 dorsal arch and C1–C2 articular surfaces)	No	19
2	LM: 2 Wi: 0	AS: 2 SP: 1	Score 0: 4 Score 1: 1 Score 2: 0	LM, AS, SP LM	Yes	No	No	20
3	LM: 2 Wi: 0	AS: 2 SP: 1	Score 0: 3 Score 1: 2 Score 2: 0	LM, AS, SP LM, AS	No	No	Yes	11
4	LM: 2 Wi: 0	AS: 2 SP: 1	Score 0: 2 Score 1: 1 Score 2: 2	AS, SP LM LM, AS	No	Yes (C1 dorsal arch)	No	12
5	LM: 2 Wi: 0	AS: 2 SP: 1	Score 0: 4 Score 1: 1 Score 2: 0	LM, AS, SP AS	No	No	No	16
6	LM: 2 Wi: 0	AS: 2 SP: 1	Score 0: 3 Score 1: 2 Score 2: 0	LM, AS, SP LM, AS	No	Partial bone remodelling (C1 dorsal arch and C1–C2 articular surfaces)	No	16
7	LM: 2 Wi: 0	AS: 2 SP: 2	Score 0: 4 Score 1: 1 Score 2: 1	AS, SP LM LM	No	No	No	10
8	LM: 2 Wi: 0	AS: 1 SP: 1	Score 0: 3 Score 1: 1 Score 2: 0	LM, AS, SP AS	No	No	No	5
9 ^2^	LM: 2 Wi: 2	AS: 2 SP: 2	Score 0: 6 Score 1: 2 Score 2: 0	LM, Wi, AS, SP LM, AS	No	Partial bone remodelling (C1 ventral arch)	No	6
10	LM: 2 Wi: 2	AS: 2 SP: 2	Score 0: 6 Score 1: 2 Score 2: 0	LM, Wi, AS, SP AS, Wi	No	n/a	n/a	n/a
11	LM: 2 Wi: 2	AS: 2 SP: 2	Score 0: 5 Score 1: 2 Score 2: 1	LM, AS, SP LM, AS AS	Yes	No	No	6
12	LM: 2 Wi: 2	AS: 2 SP: 2	Score 0: 7 Score 1: 1 Score 2: 0	LM, Wi, AS, SP LM	Yes	Partial bone remodelling (C1–C2 right articular surface)	No	6

^1^ Screw placement score: 0, optimal; 1, suboptimal; 2, hazardous; 3, dangerous. LM: C1 lateral masses; Wi: C1 wings; AS: C2 cranial articular surfaces; SP: C2 spinous process; n/a: not available. ^2^ This dog also had 5 optimally placed self-drilling self-tapping monocortical titanium screws placed to anchor a titanium mesh.

## Data Availability

Not applicable.

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
