# Peer review of "Evaluation of a Novel Dorsal-Cemented Technique for Atlantoaxial Stabilisation in 12 Dogs"

_life, 2021, doi:10.3390/life11101039_

Round 1

Reviewer 1 Report

Authors present a retrospective study of rigid cemented dorsal atlantoaxial stabilisation (DAAS) using additional bone corridors than previously reported in 12 consecutive dogs. The method involved bi-cortical screws placed in at least 4 of 8 available bone corridors, embedded in polymethylmethacrylate.  A total of 72 atlantoaxial screws were placed, 51 (70.8%) were optimal, 17 (23.6%) were suboptimal and 4 (5.6%) were graded as hazardous.  The clinical outcome was considered good to excellent in all but one case displaying episodic discomfort despite appropriate atlantoaxial reduction with a single construct failure. This study suggests the proposed DAAS is a viable alternative to ventral techniques. 

This is an innovative study which present interesting applications of spinal navigation for dorsal stabilization of the cervical spine and the results of this study could find use also in human cadaver experimental research. Limitations are retrospective character of the study and the different types of dogs being used. Duration of the surgery should be included in the tables for each dog. Please add a CT scan of optimal, suboptimal and hazardous screw. It would be useful to describe in detail the surgical workflow; was the spinal navigation used direct or indirect (only using the 3D model?). Authors mention the planned screw trajectories; but it is unclear if this trajectories were used for navigated screw placement and were planned at the actual operative situs or do you refer to trajectories on the 3D model? If applicable please include at least one representative case with preoperative and postoperative imaging, including MRI when possible. What was the overall fusion rate to follow up? Is it possible to measure and report the mean deviation of the entry point as well as the tip and angular deviation of the implanted screws compared to the planned screws?

Author Response

Mr. Aleksa Subonj

Editor, Assistant Editor, MDPI Novi Sad

26th September 2021

Re: Resubmission of manuscript ID of life-1372262

Title: " Evaluation of a novel dorsal cemented technique for atlantoaxial stabilisation in 12 dogs"

Authors: Joana Tabanez, Rodrigo Gutierrez-Quintana, Adriana Kaczmarska, Roberto José-López, Veronica Gonzalo Nadal, Carina Rotter, Guillaume Leblond

Dear Mr. Subonj,

We thank you for considering the above manuscript for publication. We extend our thanks to the reviewers and editorial board for the time they have invested, the constructive feedback and considerable efforts to improve the quality of our manuscript. We have addressed the points raised below. Changes are illustrated in blue and bold and can be tracked in the word document. Please see the attachments.

Reviewer 1 suggestions and corrections:

“Authors present a retrospective study of rigid cemented dorsal atlantoaxial stabilisation (DAAS) using additional bone corridors than previously reported in 12 consecutive dogs. The method involved bi-cortical screws placed in at least 4 of 8 available bone corridors, embedded in polymethylmethacrylate.  A total of 72 atlantoaxial screws were placed, 51 (70.8%) were optimal, 17 (23.6%) were suboptimal and 4 (5.6%) were graded as hazardous.  The clinical outcome was considered good to excellent in all but one case displaying episodic discomfort despite appropriate atlantoaxial reduction with a single construct failure. This study suggests the proposed DAAS is a viable alternative to ventral techniques.”

“This is an innovative study which present interesting applications of spinal navigation for dorsal stabilization of the cervical spine and the results of this study could find use also in human cadaver experimental research. Limitations are retrospective character of the study and the different types of dogs being used. “

  1. “Duration of the surgery should be included in the tables for each dog.”

Reply: thank you for your suggestion, the duration of the surgery for each dog has been added to table 1.
A phrase was added on line 263: “Median surgery time was 207 minutes (range 155-300 min).”

  1. “Please add a CT scan of optimal, suboptimal and hazardous screw. “

Reply: We thank the reviewer for this suggestion. A figure number 6 has been added on line 361 with examples of optimal, suboptimal and hazardous screws in a single case. It is difficult to display the relationship between the planned screw and the actual screw position in a single plane (because they are not located in one plane), therefore we propose superimposing screw models to the CT transverse image for better understanding (see below and attachments).

SEE FIGURE 6 attached

Figure 6. MR and CT images with superimposed 3D models depicting C1-C2 reduction and screw accuracy obtained in case 7. (a) Preoperative sagittal MR image with dorsal displacement of C2 dens (white arrowhead); (b) Postoperative sagittal MR image demonstrating C1-C2 reduction and the benefit of using titanium screws allowing visualisation of the local anatomy; (c) Dorsal 3D reconstruction depicting the overall screw placement accuracy; (d) Cranial view of CT transverse image illustrating a hazardous position of the left lateral mass screw due to slight vertebral canal violation (dashed arrow); (e) Caudal view of a CT transverse image and C1 3D model revealing the suboptimal position of the right lateral mass screw due to penetration of the C1-C2 synovial joint (white arrow); (f) Cranial view of a CT transverse image and C2 3D model depicting 3 optimal screw placements (bicortical within the intended bone corridor). Yellow screws: preoperative planned positions; Blue screws: actual postoperative positions.

A phrase was added on line 328: “A case example is presented in Figure 6 depicting C1-C2 reduction and screw accuracy.”

  1. “It would be useful to describe in detail the surgical workflow; was the spinal navigation used direct or indirect (only using the 3D model?).”

Reply: Thank you for highlighting the confusion in our method. The methodology is not a proper spinal navigation system. Screws were positioned using precalculated inclination angles, drilling lengths and visual approximations comparing 3D models on a computer screen and the actual patient’s anatomy revealed by the surgical approach. Visual approximation was achieved using the video recordings of the surgical plan (Video S1).

We have modified the following phrase on line number 141: “Drilling direction was either estimated by visual assessment of a video recording depicting 3D screw positions on a computer display or using a wedge osteotomy gauge to match the calculated values of inclination angles to the sagittal plane (Figure 2c).

  1. “Authors mention the planned screw trajectories; but it is unclear if this trajectories were used for navigated screw placement and were planned at the actual operative situs or do you refer to trajectories on the 3D model?”

Reply: We apologise for the confusion. As per point 3, the screw trajectories were initially planned on 3D models based on preoperative CT images. These planned trajectories were the basis of the intraoperative guidance, but we did not use a neuronavigational software or precise guides to match the planned trajectory to the actual patient’s anatomy. Instead, we used 3D visual representations on a computer display available intraoperatively. The insertion drilling points were determined visually while trajectories were either visually determined or using wedge osteotomy gauge to help the surgeon matching calculated values. We hope the addition from point 3 is enough to clarify the methodology.

  1. If applicable please include at least one representative case with preoperative and postoperative imaging, including MRI when possible.

Reply: We thank the reviewer for this suggestion. Figure number 6 should also cover this point.

  1. What was the overall fusion rate to follow up?

Reply: We thank the reviewer for this suggestion.

We have added a phrase on line 446: “
Overall, continuous bone fusion was observed in 2 dogs (16.6%) and partial bone fusion (with residual separating line) in 3 dogs (25%).”

  1. Is it possible to measure and report the mean deviation of the entry point as well as the tip and angular deviation of the implanted screws compared to the planned screws?

Reply: We thank the reviewer for this suggestion. While we certainly understand the value of objective outcome measures to assess implant placement accuracy, this was not the focus of our study. Our aim was to present the surgical technique while demonstrating safety and comparing implant placement accuracy to similar data in the literature. We decided to use a semi-quantitative method by assessing the amount of breach outside the intended corridor because it provides direct clinical value and because previous studies on the subject have used similar grading systems.
We do agree that deviation of entry point and trajectories would be better to assess the accuracy of our proposed planning method however this would require a prospective study design with properly controlled methodology and was not the study aim. The presented case series contains some intrinsic variability due to its retrospective nature and the learning curve of the surgeons who were performing the technique for the first time initially. The retrospective nature of the study also implies that surgeons used the planning method at their discretion and some preferred using numerical values while othered preferred visual assessments. Therefore, we believe that the screw scoring system we have selected is more relevant to achieve the study aim (assessing clinical safety) and would prefer not to complicate any further the manuscript with additional data.

Additional Notes:
Based on comments from both reviewers about spell checking, we have asked a British colleague to review the manuscript. Suggested changes have been included to the submitted version with tracked changes applied.

We have added a credit omitted in the initial submitted manuscript on line 650: We thank Dr. Bethan Jones for helping in the gathering of the photographs presented in Figure 2.

Reviewer 2 Report

Overall this is a well written article with appropriate follow up on a novel posterior stabilization technique for canines with AA instability.  Limitations are the retrospective nature of the paper, however, given the paucity of literature on this technique and the good outcomes, I don't feel this severely limits the paper.

Author Response

Mr. Aleksa Subonj

Editor, Assistant Editor, MDPI Novi Sad

26th September 2021

Re: Resubmission of manuscript ID of life-1372262

Title: " Evaluation of a novel dorsal cemented technique for atlantoaxial stabilisation in 12 dogs"

Authors: Joana Tabanez, Rodrigo Gutierrez-Quintana, Adriana Kaczmarska, Roberto José-López, Veronica Gonzalo Nadal, Carina Rotter, Guillaume Leblond

Dear Mr. Subonj,

We thank you for considering the above manuscript for publication. We extend our thanks to the reviewers and editorial board for the time they have invested, the constructive feedback and considerable efforts to improve the quality of our manuscript. We have addressed the points raised below. Changes are illustrated in blue and bold and can be tracked in the word document.

Reviewer 2 comments:

“Authors present a retrospective study of rigid cemented dorsal atlantoaxial stabilisation (DAAS) using additional bone Overall this is a well written article with appropriate follow up on a novel posterior stabilization technique for canines with AA instability.  Limitations are the retrospective nature of the paper, however, given the paucity of literature on this technique and the good outcomes, I don't feel this severely limits the paper.”

Reply: We thank the reviewer for these comments and recognise the main limitation of this study.

Additional Notes:
Based on comments from both reviewers about spell checking, we have asked a British colleague to review the manuscript. Suggested changes have been included to the submitted version with tracked changes applied.

We have added a credit omitted in the initial submitted manuscript on line 650: We thank Dr. Bethan Jones for helping in the gathering of the photographs presented in Figure 2

Round 2

Reviewer 1 Report

The authors have sufficiently answered the requests; please include link for the video or the video as a supplemental material.